# Host-guest charge transfer for scalable single crystal epitaxy of a metal-organic framework
Arthur Mantel[1], Berthold Stöger [2], Alexander Prado-Roller[3] & Hidetsugu Shiozawa [1,4] ✉

Methods to grow large crystals provide the foundation for material science and technology. Here we demonstrate single crystal homoepitaxy of a metal-organic framework (MOF) built of zinc, acetate and terephthalate ions, that encapsulate arrays of octahedral zinc dimethyl sulfoxide (DMSO) complex cations within its one-dimensional (1D) channels. The three-dimensional framework is built of two-dimensional Zn-terephthalate square lattices interconnected by anionic acetate pillars through diatomic zinc nodes. The charge of the anionic framework is neutralized by the 1D arrays of $Zn(DMSO)_6^{2+}$ cations that fill every second 1D channel of the framework. It is demonstrated that the repeatable and scalable epitaxy allows square cuboids of this charge-transfer MOF to grow stepwise to sizes in the centimeter range. The continuous growth with no size limits can be attributed to the ionic nature of the anionic framework with cationic 1D molecular fillers. These findings pave the way for epitaxial growth of bulk crystals of MOFs.

Metal-organic frameworks (MOFs) are a class of porous materials that are considered to be used in a wide range of applications[1,2]. Syntheses of high quality and bulk single crystals are elemental for fundamental research and applications of MOFs[3,4]. The dimensions of some of the well-studied MOFs can get as large as $3 \times 3 \times 2$ mm for MOF-5[5,6], and $2 \times 3 \times 4$ mm for HKUST-1[7]. The size of MOF crystals reported thus far reaches 6–7 mm in length, as achieved for Rb and Cs cyclodextrine MOFs (CD-MOF-2 (Rb) and CD-MOF-3)[8] based on vapor diffusion and reseeding, as well as for [Zn(3-ptz)2]n (MIRO-101) with acid modulators[9]. More recently, exploiting nucleation and growth environment provided by the Marangoni effect allowed centimeter-long thin needles of Zn-based MOF crystals to be synthesized[10]. However, bulk crystal growth of MOFs is yet to be achieved for both fundamental sciences[11] and practical applications of MOFs.

In this study, we synthesize a MOF built of zinc, acetate, and terephthalate ions, that encapsulate arrays of an octahedral zinc dimethyl sulfoxide (DMSO) complex within its one-dimensional (1D) channels, by mixing zinc acetate and terephthalic acid (TPA) in DMSO, a rarely used solvent for the synthesis of MOFs. Zinc acetate and TPA in a more common solvent, such as N,N-diethylformamide (DEF) or N,N-dimethylformamide (DMF), are the well-known precursors for the synthesis of MOF-5. It is demonstrated that these precursors in DMSO lead to the formation of large crystals. Single crystal X-ray diffraction (SXRD) combined with density-functional theory (DFT) calculations reveals the three-dimensional nano

architecture made of two-dimensional (2D) Zn-terephthalate square lattices interconnected by anionic acetate pillars through diatomic zinc nodes. The negative charge of the framework is neutralized by the 1D arrays of $Zn(DMSO)_6^{2+}$ cations that fill every second 1D channel of the framework. Furthermore, we demonstrate that the stepwise homoepitaxial growth allows square cuboids of this charge-transfer MOF to grow as large as $19.8 \times 4.2 \times 4.2$ mm. The continuous growth with no size limits can be attributed to host-guest electrostatic attraction between the anionic framework and the cationic 1D fillers, that stabilizes the structure at a molecular level.

## Results and discussion
### First crystal growth
We choose DMSO as a solvent. TPA or 1,4-benzenedicarboxylic acid has a relatively high solubility of 20 wt% in DMSO[12]. This provides a wider range of concentrations as compared with dimethylformamide (the solubility of TPA is 6.7 wt%) that is widely used for the synthesis of MOFs. Specifically, MOF-5 is known to be synthesized with TPA in N,N-diethylformamide (DEF)[5,6] and in N,N-dimethylformamide (DMF)[13,14]. DMSO has been rarely used as a solvent[15], or a co-solvent[16], for the synthesis of MOFs.

Crystals of the charge-transfer MOF are synthesized by mixing zinc acetate (ZnOAc) and TPA in DMSO followed by heating at 40 °C. For more details see the Methods section. Figure 1 displays the photographs of crystals

[1]J. Heyrovsky Institute of Physical Chemistry, Czech Academy of Sciences, Prague, Czechia. [2]X-ray Centre, TU Wien, Vienna, Austria. [3]Department of Inorganic Chemistry, University of Vienna, Vienna, Austria. [4]Faculty of Physics, University of Vienna, Vienna, Austria. ✉e-mail: hidetsugu.shiozawa@univie.ac.at

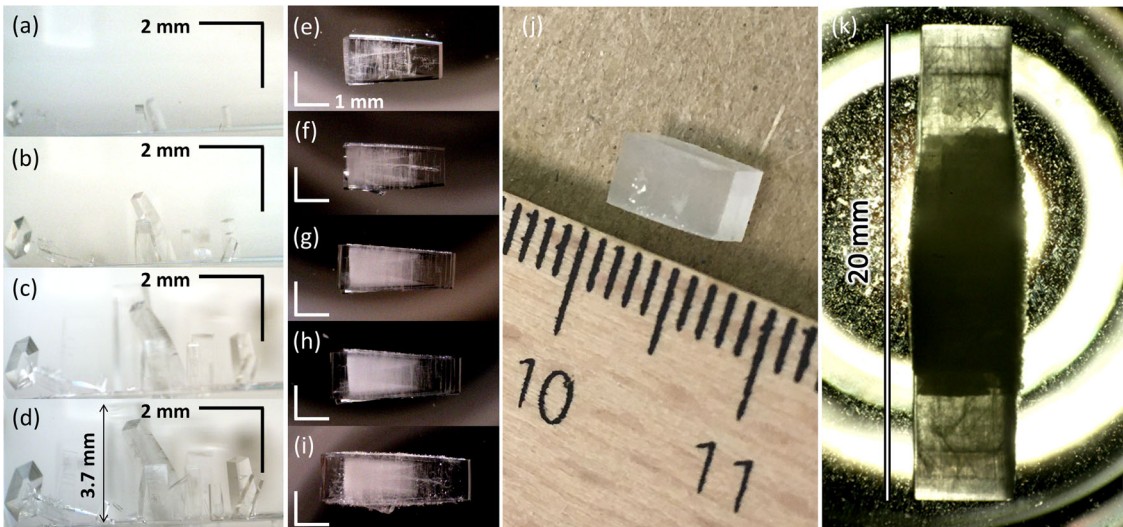

**Fig. 1 | Photographs of crystals.** Photographs of crystals growing in the original precursor solution, taken after (**a**) 6 days (**b**) 8 days (**c**) 10 days (**d**) 13 days of the first reaction. The tallest crystal reaches 3.7 mm in length. Photographs of the crystal in the secondary precursor solution, taken after (**e**) 0 day (**f**) 4 days (**g**) 7 days (**h**) 11 days (**i**) 13 days of the multiple reaction steps. The reaction medium was replaced by a fresh one just before taking each photograph. **j** The cuboid with a length of ~6 mm cut from a larger crystal. **k** The same cuboid after multiple reaction steps over five months in a glass vial with a diameter of 27.5 mm which is back-illuminated. The crystal dimensions are 19.8 × 4.2 × 4.2 mm.

**Fig. 2 | Crystal structure from SXRD.** Crystal structure of the charge-transfer MOF viewed along the a axis (**a**), the b axis (**b**) and the c axis (**c**), and an arbitrary direction (**d**).

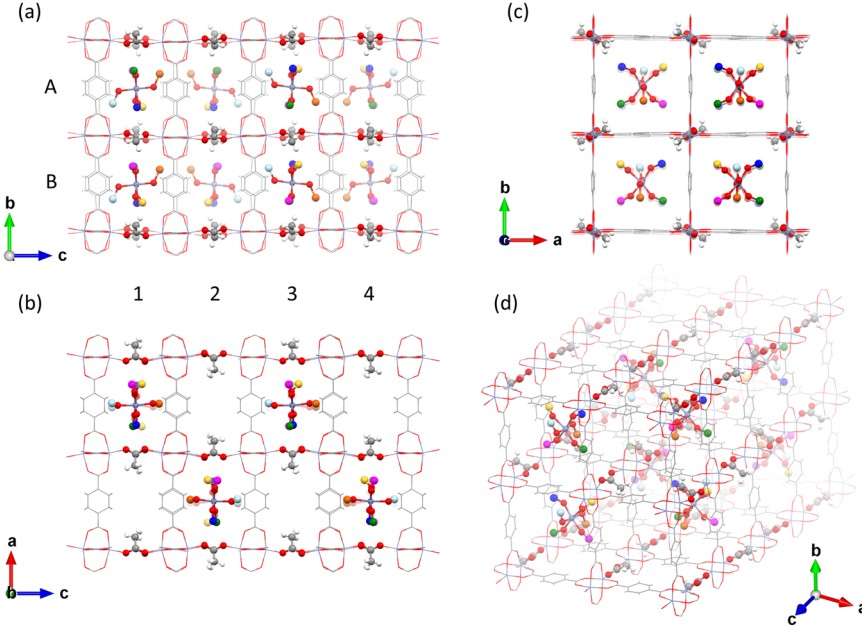

grown at the bottom of the glass vial in the precursor solution, taken after (a) 6 days (b) 8 days (c) 10 days, and (d) 13 days of the reaction. In 13 days, transparent square cuboid crystals grow as long as 3.7 mm.

## Crystal structure

The crystal structure has been studied using SXRD combined with DFT calculations. For experimental details see Supplementary Information, section S1. Figure 2 shows the unit cell of the charge-transfer MOF viewed along the a axis (a), the b axis (b) and the c axis (c), and an arbitrary direction (d). The MOF crystallizes in an orthorhombic structure with a space group of *Pbca*, lattice constants of $a = 21.8209(18)$, $b = 21.8491(11)$, $c = 35.597(3)$ Å, and axial angles $\alpha = \beta = \gamma = 90°$. The unit cell is composed of $2 \times 2 \times 4 = 16$ cuboidal subunits. The framework is built of the 2D lattices

on the a-b plane and the pillar units along the c axis. The 2D lattice depicted by wire frames is composed of zinc dimer and terephthalate or 1,4-benzenedicarboxylate (BDC) linkers (see panels a and b). The terephthalate's molecular plane is perpendicular to the 2D lattice plane. These 2D lattices are pillared by acetate ions (depicted in the ball and stick model) to construct the three-dimensional framework with the 1D channels along the b axis. The paddle wheel secondary building blocks (SBUs) composed of Zn dimer, four terephthalates, and two acetate ions are visible in panel a and b.

Importantly, the SXRD elucidates that every second channel along the b axis is filled with a chain of $Zn(SO)_6$ molecules. The cif file is available as Supplementary Data 1. In other words, every other channel ensures the open porosity. There are eight different $Zn(SO)_6$

molecules in the unit cell, which makes the length of the unit cell along the c axis, as long as the four cuboidal subcells, while the lattice constants along the a and b axes span two cuboidal subcells. Every second 1D void along the b axis encapsulates a chain of $Zn(SO)_6$, as recognized in Fig. 2b.

In Fig. 2, six sulfur atoms of each $Zn(SO)_6$ are depicted in different colors. In panel a, all eight molecules are visble. With letters A and B, and numbers 1, 2, 3, and 4, all eight molecules are labeled as 1A, 1B, 2A, 2B, 3A, 3B, 4A, and 4B. Molecules 1A, 1B, 3A, and 3B are in the front cuboids, and molecules 2A, 2B, 4A, and 4B are in the cuboids behind. The depth is expressed as increased transparency.

The nearest-neighbor $Zn(SO)_6$ molecules, namely A and B, are along the 1D chain, and their structures are mirrored from each others' against the b-c plane. The second nearest-neighbor $Zn(SO)_6$ molecules are those in the nearest-neighbor chains in the same a-c plane, namely #1 and #2, and #3 and #4, and, and their structures are mirrored from each others' against the a-b plane. The structure of the molecules in #1 and #3 (#2 and #4) in the same a-c plane are mirrored from each others' against the a-c plane. These three reflection operations allow all eight molecular orientations to be reproduced from one orientation.

The detailed molecular structure of the filler could not be identified by the SXRD because of disorder. Under the synthesis conditions, $Zn(DMSO)_6^{2+}$ are most likely to be present in the precursor solution. The molecular formula of the MOF is then $(Zn_2(BDC)_2OAc^-)_2Zn(DMSO)_6^{2+}$. In order to optimize the molecular structure of $Zn(DMSO)_6^{2+}$ in the framework, the DFT calculations have been carried out in one of the 16 cuboidal subunits. Other seven orientations of $Zn(DMSO)_6^{2+}$ in the unit cell can be reproduced using the three reflection operations discussed above. The optimized structures of $Zn(DMSO)_6^{2+}$ in two adjacent subcells are depicted in Fig. 3a. It shows the orientations of the Zn-S bonds similar to those for the experimentally identified $Zn(SO)_6$ in Fig. 2c. See all eight molecules in the unit cell in Supplementary Information, section S1.3, Fig. S3.

### Electrostatic potential

Each $Zn(DMSO)_6^{2+}$ filler is paired with two of acetate anion $OAc^-$ with their carboxylic groups facing the filler, as shown in Fig. 3a. As a result, acetate ions alternate their orientations along the c axis. In Fig. 3a, two adjacent pairs of $Zn(DMSO)_6^{2+}$ and acetate anions in the 1D chain along the b axis are shown. The deprotonated carboxyl group of the acetate anion faces the $Zn(DMSO)_6^{2+}$ to compensate the charge, which leads to the zizgag orientations of the acetate anions. The methyl groups protrude into the next 1D channel along the b axis, which leaves every second 1D channel unfilled.

The corresponding calculated electrostatic potential map is shown in Fig. 3b. It demonstrates perfectly the host-guest ionic pair of opposite charges.

### Ionic MOFs

MOFs can encapsulate foreign atoms, molecules, and nanoparticles inside the pores. The process of the encapsulation of guest materials can be in-situ (during the synthesis of MOFs)[17–19] or post-synthesis (after the synthesis of neutral MOFs)[20–24]. In the latter case, the MOF can be anionic or cationic, depending on the ionization potential/electron affinity of the guest species and of the ligand[25]. The in-situ method requires ionic secondary building units to build ionic MOFs counterposed by ionic fillers of opposite charge. The hosting MOFs can be cationic (positively charged)[18,26–33], or anionic (negatively charged)[34–39]. Among them, anionic MOFs reported in refs. 38,39 are built of anionic pillars. Anion-pillared MOFs are formed by the combination of metal nodes and organic ligands with bifunctional inorganic or organic anion pillars. Inorganic anion pillars can be linear[40] or angular[41]. Our MOF can be classified into anion-pillared MOFs with organic anion pillars[38,39].

Among MOFs built of Zn and terephthalate reported thus far, e.g., $Zn(BDC) \cdot (DMF)(H_2O)$[42], MOF-5 (also called IRMOF-1)[13], MOF-2[43], and others[44–46], a family of MOFs in which 2D layers consisting of linear trinuclear zinc and terephthalate are pillared by neutral $Zn(O2CR)6$, formate homoanions, or terephthalate dianions bear certain resemblance to ours[45].

### Homoepitaxy

Now we demonstrate homoepitaxy on the crystal grown in the first step as a seed whose size, as shown in Fig. 1d, is already comparable to some of the largest bulk MOF crystals reported thus far[8,9]. Figure 1e–i are the photographs of a crystal in the secondary precursor solution, taken after (e) 0 day (f) 4 days (g) 7 days (h) 11 days (i) 13 days of the multiple reaction steps. The reaction medium was replaced by a fresh one just before taking each photograph. In order to avoid nucleation of small crystals, the molar ratio of Zn to terephthalate for the secondary solution was adjusted in the range from 4:1 to 5:1, as well as the solution was replaced to a fresh one within 2–4 days. For more details see the Methods section and Supplementary Information, section S2. It is demonstrated that the crystal grows further while maintaining the shape in a homoepitaxial manner. The growth into two opposite directions is faster so freshly extended sections are obvious in the photographs.

Figure 1k is the photograph of a MOF crystal grown stepwise for about five months. The dimensions of the square cuboid crystal reaches ~19.8 × 4.2 × 4.2 mm and it continues to grow further especially at both top and bottom ends. Note that the crystal gets less transparent in the middle old section, which can be attributed to aging. Also, small crystals grow in the middle section of the crystal surface, which need to be scraped off in order for the crystal to epitaxially grow on the side wall, while the top and bottom end surfaces stay clean so the crystal continues to extend its length in a freshly prepared precursor solution.

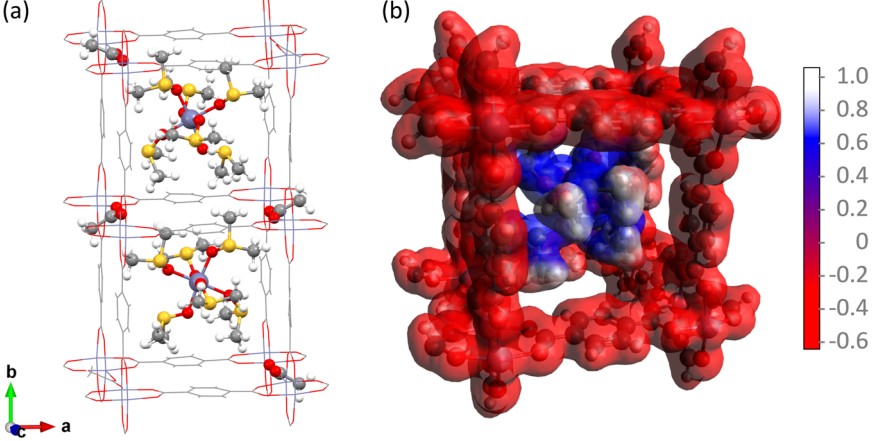

**Fig. 3 | Molecular structure and electrostatic potential map from DFT. a** Two adjacent pairs of $Zn(DMSO)_6^{2+}$ and two acetate anions in the 1D chain along the b axis. **b** Electrostatic potential map of a $Zn(DMSO)_6$ in a cuboidal subcell.

**Fig. 4 | Sequential images visualizing the epitaxial growth on the a-b plane. 1** a 2D Zn-terephthalate lattice. **2** deposition of $Zn(DMSO)_6^{2+}$ (blue spheres) and acetate anions (red cylinders). **3** deposition of another 2D Zn-terephthalate lattice. **4** deposition of $Zn(DMSO)_6^{2+}$ and acetate anions.

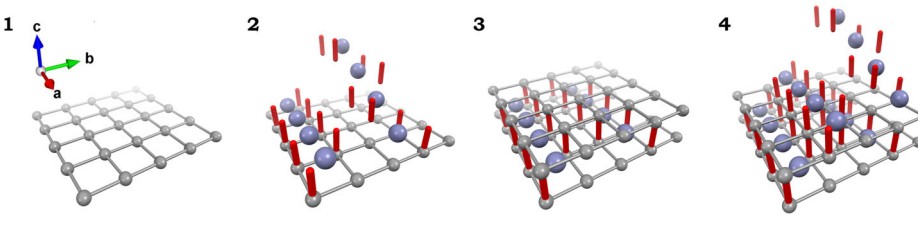

SXRD on an as-grown microcrystal reveals that the top and bottom crystal surfaces are along the 2D layer of Zn-terephthalate in the a-b plane in Fig. 2c. Along the c axis, the crystal grows by alternating depositions of the 2D Zn-terephthalate layer, acetate anion $OAc^-$ pillars, and parallel arrays of the 1D $Zn(DMSO)_6^{2+}$ chains, as illustrated in Fig. 4. The key to the successful epitaxial growth is the ionic components of the opposite charge. Namely, the negatively charged framework and the positively charged 1D molecular chains attract each other to favor the single crystal epitaxy in the same way as ionic crystals grow large. Furthermore, the charge compensation between two monoanionic pillars and one dicationic filler ensures the open porosity of every other 1D channel in the MOF. This work shows the potential of ionic ligand-guest pairs for single crystal homoepitaxy of MOFs.

## Conclusions
We have demonstrated the homoepitaxy of an anionic $(Zn_2(BDC)_2OAc^-)_2$ MOF filled with 1D arrays of $Zn(DMSO)_6^{2+}$ cations. The bulk crystal synthesized over five months extends as long as 2 cm. A large number of combinations of metal acetates and linear dicarboxylic acids in DMSO would not only allow homoepitaxy of a family of the charge-transfer MOFs, but also allow heteroepitaxy of their combinations. The single crystal epitaxy with no size limit driven by the host-guest charge transfer offers an avenue for the growth of bulk crystals of MOFs, and possibly heteroepitaxy with patterned linkers and metal species.

## Methods
### Reagents
DMSO (ACS reagent, ≥99.9%, Cat # 472301), TPA (98%, Cat # 185361), zinc acetate dehydrate (puriss. p.a., ACS reagent, ≥99.0% (KT), Cat # 96459) were purchased from Sigma-Aldrich and used without purification.

### Procedure for the first crystal growth
Zinc acetate (ZnOAc) dihydrate (2.14 g, 9.75 mmol) was dissolved in 6 mL of DMSO. TPA (0.54 g, 3.25 mmol) was dissolved in 6 mL of DMSO. The TPA solution was added to the ZnOAc solution and the mixture was stirred for 1 min. The molar ratio of ZnOAc to TPA was 3:1. 2 mL of the mixed solution was pipetted into a 20 mL glass vial with a diameter of 27 mm, then placed in an oven set at 40 °C for the first crystal growth.

### Procedure for the homoepitaxy
A crystal grown in the first step was placed in the secondary precursor solution in a glass vial of 20 mL volume and 27 mm diameter. In order to avoid nucleation of small crystals, the molar ratio of Zn to TPA for the secondary solution has been adjusted in the range from 4:1 to 5:1. The solution was replaced to a fresh one after 2–4 days. Upon every solution exchange, the crystal was washed in DMSO. After applying this method, if microcrystals were observed to remain on the crystal surfaces, the crystal was dipped in 0.6 M of ZnAc in DMSO in which the crystal surfaces were edged away as observed in Supplementary Information, Fig. S8. Finally, the crystal was washed in DMSO once again. For centimeter-long crystals, microcrystals on the surfaces in the middle section of the crystal could not be removed by applying the aforementioned washing methods. In this case, microcrystals on the surfaces were scraped off using a nail file, followed by the same washing procedure.

### Density functional theory
DFT calculations were performed with the ORCA 5.0.4 quantum chemistry package using the exchange-correlation hybrid B3LYP functional and basis set 6-31G[47]. Geometry optimization was conducted for a $Zn(DMSO)_6$ molecule encapsulated in a single cuboid lattice framework. In the latter case, the framework structure was frozen for the optimization of the $Zn(DMSO)_6$ structure to save computational resources. The electrostatic potential was calculated using ORCA 5.0.4 quantum chemistry package followed by the visualization of the potential surface using Avogadro 1.2.0 program.

### Data availability
All relevant data are available from the corresponding author upon request.

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

## Acknowledgements

This research was funded in part by the Austrian Science Fund (FWF), project P30431-N36. H. S. acknowledges the financial support from the Czech Science Foundation (GACR), project 19-15217S and 22-23407S, the Austrian Federal Ministry of Education, Science and Research (BMBWF) and OeAD-GmbH through Scientific & Technological Cooperation (WTZ) program, project CZ 18/2019, and the Ministry of Education, Youth and Sports of the Czech Republic (MEYS) through V4-Japan joint research program, project 8F21010. We thank J. Köhler, S. Loyer and A. Stangl for technical assistance.

## Author contributions

A.M. carried out synthesis, optical microscopy, data curation, density-functional theory calculations, and wrote the manuscript. B.S. carried out X-ray crystallography and data curation, and wrote the manuscript. A.P.-R. performed X-ray crystallography and data curation. H.S. conceptualized and supervised the project, provided resources, carried out data curation, funding acquisition, project administration, and wrote the manuscript.

## Competing interests

The authors declare no competing interests.
