## [Peer Review File · Communications Materials]

27th May 24

Dear Prof. Shiozawa,

Thank you for submitting your manuscript, "Host-guest charge transfer for scalable single crystal epitaxy of a metal-organic framework", to Communications Materials. It has now been seen by 3 referees, whose comments are appended below. You will see that while they find your work of interest, some important points are raised. We are interested in the possibility of publishing your study in Communications Materials, but would like to consider your response to these concerns in the form of a revised manuscript.

In particular, Reviewer 3 raised strong concerns regarding the novelty of the presented studies in comparison with the previous reports. Clear differentiation of the presented work from other previous publications need to be carefully addressed with corresponding modifications in the manuscript, and especially, its introduction. In addition, there are many technical details that need to be provided as suggested by all three reviewers.

We therefore invite you to revise and resubmit your manuscript, taking into account the points raised.

When submitting your revised manuscript, please include the following:

-A response letter with a point-by-point reply to each of the referee comments and a description of changes made. Please include the complete referee report in the response letter. Please note that the response letter must be separate to the cover letter to the editors.

-A marked-up version of the manuscript with all changes to the text in a different colored font. Please do not include tracked changes or comments. Please select the file type 'Revised Manuscript - Marked Up' when uploading the manuscript file to our online system.

-A clean version of the manuscript. Please select the file type 'Article File'.

-An updated Editorial Policy checklist, uploaded as a 'Related Manuscript File' type. This checklist is to ensure your paper complies with all relevant editorial policies. If needed, please revise your manuscript in response to these points. Please note that this form is a dynamic 'smart pdf' and must therefore be downloaded and completed in Adobe Reader. Clicking this link will download a zip file containing the pdf.

In the event that your manuscript is accepted we will provide detailed guidance on our journal policies and formatting. You may however wish to ensure that the manuscript complies with our house style at this stage. See our style and formatting guide (<https://www.nature.com/documents/commsj-phys-style-formatting-guide-accept.pdf>) and checklist (<https://www.nature.com/documents/commsj-phys-style-formatting-checklist-article.pdf>) for reference.

Data availability statements and data citations policy: All Communications Materials manuscripts must include a section titled "Data Availability" at the end of the Methods section or main text (if no Methods). More information on this policy, and a list of examples, is available at <http://www.nature.com/authors/policies/data/data-availability-statements-data-citations.pdf>.

- Accession codes for deposited data
- Other unique identifiers (such as DOIs and hyperlinks for any other datasets)
- At a minimum, a statement confirming that all relevant data are available from the authors
- If applicable, a statement regarding data available with restrictions
- If a dataset has a Digital Object Identifier (DOI) as its unique identifier, we strongly encourage including this in the Reference list and citing the dataset in the Data Availability Statement.

DATA SOURCES: We strongly encourage authors to deposit all new data associated with the paper in a persistent repository where they can be freely and enduringly accessed. We recommend submitting the data to discipline-specific, community-recognized repositories, where possible and a list of recommended repositories is provided at <http://www.nature.com/sdata/policies/repositories>.

If a community resource is unavailable, data can be submitted to generalist repositories such as figshare or Dryad Digital Repository. Please provide a unique identifier for the data (for example a DOI or a permanent URL) in the data availability statement, if possible. If the repository does not provide identifiers, we encourage authors to supply the search terms that will return the data. For

data that have been obtained from publically available sources, please provide a URL and the specific data product name in the data availability statement. Data with a DOI should be further cited in the methods reference section.

Please use the following link to submit your documents:

[link redacted]

We hope to receive your revised paper within three months; please let us know if you aren't able to submit it within this time so that we can discuss how best to proceed. If we don't hear from you, and the revision process takes significantly longer, we will close your file. In this event, we will still be happy to reconsider your paper at a later date, as long as nothing similar has been accepted for publication at Communications Materials or published elsewhere in the meantime.

Please do not hesitate to contact me if you have any questions or would like to discuss these revisions further. We look forward to seeing the revised manuscript and thank you for the opportunity to review your work.

Best regards,

Natalia Shustova, Prof.

Editorial Board Member

Communications Materials

orcid.org/0000-0003-3952-1949

Reviewers' comments:

Reviewer #1 (Remarks to the Author):

This paper demonstrates single crystal homoepitaxy of a metal-organic framework (MOF) built of zinc, acetate and terephthalate ions, that encapsulate arrays of octahedral zinc dimethyl sulfoxide (DMSO) complex cations within its one-dimensional (1D) channels. The continuous growth with no size limits can be provided a reference method for epitaxial growth of bulky crystals of MOFs. However, due to the confusion in the article's logical writing, a series of revisions should be made before publication.

Here are some reasons:

1. The logic in the supporting information should match the order of the main text, and when quoting content from the supporting information, please indicate the specific chapter or section being referenced. For example, "Crystals of the MOF are synthesized by mixing Zinc acetate (ZnOAc) and TPA in DMSO followed by heating at 40 °C. For more details see the S1." can be changed to "For more details see the S3".
2. Please avoid using ambiguous words in describing the details of experimental steps, such as "a few days later", and What are the reasons why DMSO is rarely used in MOF synthesis? Other researchers often choose DMF or DEF as solvents. I hope you can provide explanations for the advantages of using DMSO and enrich the literature support.
3. In Section 2.3, the electrostatic potential of the MOF material was simulated and calculated, confirming the opposite charges of the host-guest ion pairs. If there are relevant references, I hope you can supplement them.
4. The logic of the images in the main text is confusing. I hope the author can reorganize the writing logic of the article to ensure that the textual content, images, and supporting information mutually support each other in terms of logic.

Reviewer #2 (Remarks to the Author):

The method to grow high-quality large crystals has been a major challenge in MOF synthesis. In this work, homoepitaxy MOF-5 centimeter-sized bulk crystals were synthesized using zinc acetate and terephthalic acid in DMSO solvent. Single crystal X-ray diffraction (SXRD) combined with density-functional theory (DFT) calculations reveals that the three-dimensional framework is built of two-dimensional Zn-terephthalate square lattices interconnected by anionic acetate pillars through diatomic zinc nodes. More importantly, electrostatic interactions between the encapsulated in 1D arrays of Zn(DMSO)₆²⁺ cations and the host framework stabilize the framework structure, which is

key to achieving repeatable and scalable homoepitaxy growth. However, before it has been accepted, several raised issues should be described.

1. How to prove the homogeneous epitaxial growth of high-quality single crystals? Experimental characterization argument is inadequate. The authors also mention the aging, twinning, and small crystals on the surface that can occur with prolonged reactions.

2. The authors need to provide clear evidence of how to determine the molecular formula of the compound, please provide more detailed crystal data. CheckCIF files also should be provided, authors need to solve A and B level alerts before submission.

3. In section 2.4, the overview of ionic MOFs is too long and should focus more on discussing the interaction of guest molecules Zn(DMSO)₆²⁺ cations with the anionic acetate pillars.

4. What was the conclusion of the discussion in section S2.3 about the stability of MOF crystals in different environments?

Reviewer #3 (Remarks to the Author):

Comments to the manuscript: COMMSMAT-24-0286-T.

In this current article, “Host-guest charge transfer for scalable single crystal epitaxy of a metal-organic framework”, authors’ reported methodology to grow large crystal of MOF utilizing smaller seed crystals in a homo-epitaxial growth. The work carried out is interesting, however, it doesn’t give new insight to the studies already carried out in fabricating large crystals of MOF. Major claim in this article is the fabrication of large single crystal Zn-MOF utilizing homoepitaxial growth on a seed crystal. The structure has been confirmed using Single crystal XRD and DFT together. It has been earlier reported for synthesis of few mm sized MOF single crystals [e.g. MOF-5 (some cited by authors) and others] utilizing wet-chemical solvothermal methods for nucleation and aging [Angew. Chem. Int. Ed. 2011, 50, 276–279, J. Am. Chem. Soc. 2010, 132, 46, 16358–16361, Cell Rep. Phys. Sci. 2022; 3101004, CrystEngComm, 2013, 15, 4094-4098, CrystEngComm, 2013, 15, 4094-4098, ACS Omega 2021, 6, 27, 17289–17298]. The initial growth of the crystal seed is similar to usually reported MOF single crystal growth utilizing different parameter and solvent, however the homoepitaxial growth is interesting. In this context, the work done doesn’t provide new insights. I recommend to submit to more specific synthesis journals like Crystal growth journals. Apart from this, a few more points I would like the authors to consider which will improve the quality of manuscript:

1. Authors did not convey in the text how their synthesis and growth method is unique and different than others reported already? Why they choose Zinc salt Zn(acac)₂ in this case not Zn(NO₃)₂/ZnCl₂? As the text mostly describes crystal growth, a specific text to describe the same is sought.

2. In page 3, authors mentioned, “The continuous growth with no size limits can be attributed to host-guest electrostatic attraction between the anionic framework and the cationic 1D fillers that stabilizes the structure at a molecular level.” What is the biggest crystal size grown in the seed solution? What was the reaction time for the first crystal growth? Please mention.

3. Authors’ mentioned “DMSO has been rarely used as a solvent”. As known from literature, DMSO has strong coordinating ability with metal ions and a high boiling solvent and complicate in solvent removal process. However, in this case, it is seen that due to higher solubility of precursors and as depicted by the SCXRD structure Zn(DMSO) does stabilizes the pillar structure. Did the authors’ check for other solvent system in the similar preparation method to grow the crystals? Solvent system is very important in achieving high quality MOF crystals. If DMSO is helping the crystals to grow huge then its benefit should be highlighted in the text.

4. In the supporting information, how figure S5 and S6 are different? What was influence of air/moisture? Did the authors used the same setup for large crystal growth? Please clarify.

5. Authors mentioned, to avoid nucleation of small crystals, the molar ratio of Zn to TPA was increased from ZnOAc to TPA is (3.0 to 1.0) to 4.0 to 5.0. What was the mechanistic influence of such ratio in term of secondary structure building? Please mention in text.

6. What was size of the crystal for SC-XRD shown in figure S2b. Please mention.

7. MOF applications that rely on crystal pore engineering. Do the authors have estimate about the pore size distribution in homoepitaxially grown larger crystal structure?

8. What was the basis of the time/days taken for large crystal growth? Was it just random? Did the authors washed the crystals in between mother solution change?

9. Authors have utilized homoepitaxial growth to grow larger crystals using a secondary solution on successive interval of time, basically a multi-step nature of the crystal growth. However, the secondary growth may give rise to dislocation or interfacial defects that could be detrimental to the properties of MOF crystals in terms of application. Did the authors do any physical property measurement to check for the same?

10. For a homoepitaxial growth, I encourage authors to mention the in-plane orientations of the crystal along with the out-of-plane direction in the figure S2b.

11. It is seen in the optical images provided that the opacity of the crystal increases at the core, as the crystal grows to centimeter size (See fig 1). Authors mentioned “also, small crystals grow in the middle section of the crystal surface, which need to be scraped off in order for the crystal to epitaxially grow on the side wall”. How the authors getting rid of the smaller crystals during synthesis? Is it in-situ or they are doing on successive session manually? Please mention.

Also, for opto-electronic application, a transparent crystal is sought. How does the inner opaque core impact the application of such crystal in electronic application? Please comment.

12. How many similar large crystals authors' have grown to check the reproducibility? Do all the crystal grows with same kinetics to similar size?

Response to Reviewers' comments:

All changes made in the revised manuscript and SI are highlighted in red.

Reviewer #1 (Remarks to the Author):

This paper demonstrates single crystal homoepitaxy of a metal-organic framework (MOF) built of zinc, acetate and terephthalate ions, that encapsulate arrays of octahedral zinc dimethyl sulfoxide (DMSO) complex cations within its one-dimensional (1D) channels. The continuous growth with no size limits can be provided a reference method for epitaxial growth of bulky crystals of MOFs. However, due to the confusion in the article's logical writing, a series of revisions should be made before publication. Here are some reasons:

1. The logic in the supporting information should match the order of the main text, and when quoting content from the supporting information, please indicate the specific chapter or section being referenced. For example, "Crystals of the MOF are synthesized by mixing Zinc acetate (ZnOAc) and TPA in DMSO followed by heating at 40 °C. For more details see the SI." can be changed to "For more details see the S3".

Reply:

In the revised manuscript the SI sections are cited appropriately as suggested.

2. Please avoid using ambiguous words in describing the details of experimental steps, such as "a few days later", and What are the reasons why DMSO is rarely used in MOF synthesis? Other researchers often choose DMF or DEF as solvents. I hope you can provide explanations for the advantages of using DMSO and enrich the literature support.

Reply:

Our objective was to grow large single crystals of a MOF.

The first reason for choosing DMSO was the solubility. Terephthalic acid (TPA) is poorly soluble in organic solvents. The best solvent among those we tested was DMSO with the concentration as high as 20 g/100 ml. The second best solvent was DMF with the concentration up to 7,4 g/100 ml which is about 3 times worse than DMSO. This is already described in section 2.1.

The second reason is the kinetics. Unlike DMF or DEF, DMSO doesn't have an amino group which acts as activator of TPA to rapidly react. With DMSO, the crystal growth is slow and better controlled.

It is puzzling why DMSO was rarely used. Maybe most researchers prefer fast synthesis rather than achieving large crystal growth. This remains to be discussed in the community.

"several days" has been replaced with "2-4 days"

3. In Section 2.3, the electrostatic potential of the MOF material was simulated and calculated, confirming the opposite charges of the host-guest ion pairs. If there are relevant references, I hope you can supplement them.

Reply:

The simulation was made using Orca_vpot program which calculates the electrostatic potential at a given set of user defined points. The procedure is well described in the Orca manual, Ref [1] (p. 1149).

[1] F. Neese, F. Wennmohs, D. Ganyushin, M. Garcia, Y. Guo, A. Hansen, B. Helmich-paris, L. Huntington, Orca 5.0.4, 2022. <https://orcaforum.kofo.mpg.de/app.php/portal>.

In the revised manuscript, section S1.4, this is explained and the reference is cited.

4. The logic of the images in the main text is confusing. I hope the author can reorganize the writing logic of the article to ensure that the textual content, images, and supporting information mutually support each other in terms of logic.

Reply:

Thank you for pointing this out. In the revised manuscript, Figure panels 1a-d are cited as they appear first in the text as follows.

“Panels a, b, c and d in Figure 1 are photographs of crystals growing in the precursor solution taken sequentially after (a) 6 days (b) 8 days (c) 10 days (d) 13 days of the reaction. The tallest crystal reaches 3.7 mm in length.”

“, as clearly visible in Figure 2a, b and c” in section 2,3, removed as it is confusing.

Reviewer #2 (Remarks to the Author):

The method to grow high-quality large crystals has been a major challenge in MOF synthesis. In this work, homoepitaxy MOF-5 centimeter-sized bulk crystals were synthesized using zinc acetate and terephthalic acid in DMSO solvent. Single crystal X-ray diffraction (SXRD) combined with density-functional theory (DFT) calculations reveals that the three-dimensional framework is built of two-dimensional Zn-terephthalate square lattices interconnected by anionic acetate pillars through diatomic zinc nodes. More importantly, electrostatic interactions between the encapsulated in 1D arrays of $\text{Zn}(\text{DMSO})_6^{2+}$ cations and the host framework stabilize the framework structure, which is key to achieving repeatable and scalable homoepitaxy growth. However, before it has been accepted, several raised issues should be described.

1. How to prove the homogeneous epitaxial growth of high-quality single crystals? Experimental characterization argument is inadequate. The authors also mention the aging, twinning, and small crystals on the surface that can occur with prolonged reactions.

Reply:

The epitaxial growth of a single crystal of the MOF is demonstrated by its sequential growth as visualized in Figure 1 a-d for crystals in the first reaction medium, and panels e-i for a crystal in the secondary reaction medium, and in Figure S1 a-e for the multiple reaction steps for 146 days. From these observations, the crystal quality is high enough to sustain growth directions of a crystal. Twining is so minor that it doesn't result in intergrowth of multiple separate crystals. Microcrystals grow only on the surface of a large crystal perpendicular to the fast growth directions, so it does not passivate the fast epitaxial growth along the c axis. Also, they can easily be removed for further epitaxial growth on the surface perpendicular to the fast growth direction.

2. The authors need to provide clear evidence of how to determine the molecular formula of the compound, please provide more detailed crystal data. Checkcif files also should be provided, authors need to solve A and B level alerts before submission.

Reply:

The Checkcif file is attached upon the resubmission. As commented in the Checkcif file, it is common for MOFs that the crystal structure is poorly defined because of disorders in the cavities. Only weak intensities were collected and additionally the crystal was twinned.

All A- and B-level alerts in the CheckCIF report are due to disorder and weak diffraction intensities and have been commented on.

In the pores the $[\text{Zn}(\text{DMSO})_6]$ moieties could unambiguously be identified in difference Fourier maps. Moreover, S atoms of uncoordinated DMSO molecules were identified. However, the electron densities of these molecules were so diffuse that we preferred to remove them with SQUEEZE.

The revised CIF file which includes the comments to the alerts is attached.

3. In section 2.4, the overview of ionic MOFs is too long and should focus more on discussing the interaction of guest molecules $Zn(DMSO)_6^{2+}$ cations with the anionic acetate pillars.

Reply:

In section 2.4, two sentences introducing “Pillar-layered MOFs” have been deleted because of their insignificance to the discussion on the guest-host interaction.

4. What was the conclusion of the discussion in section S2.3 about the stability of MOF crystals in different environments?

Reply:

The following sentence has been added at the end of the section.

“To conclude, the MOF is stable in the mother solution sealed hermetically. DMSO is the best solvent to store the MOF when the solvent is not hermetically sealed.”

In addition, a DMSO solution of zinc acetate (ZnOAc) or terephthalic acid (TAP) dissolves MOF crystals. The dissolution in the TAP solution is much faster than in the ZnOAc solution. The latter was used to clean the crystal surfaces upon solution exchange.

In order to avoid confusion, the figures are placed in the following order.

Figure S4: Mother solution sealed (stable)

Figure S5: Mother solution not sealed (fairly stable)

Figure S6: DMSO not sealed (best among not sealed)

Figure S7: ZnOAc in DMSO not sealed (fairly stable)

Figure S8: TPA in DMSO not sealed. (not stable)

“The solution was exposed to air.” has been inserted wherever appropriate.

Reviewer #3 (Remarks to the Author):

Comments to the manuscript: COMMSMAT-24-0286-T.

In this current article, “Host-guest charge transfer for scalable single crystal epitaxy of a metal-organic framework”, authors’ reported methodology to grow large crystal of MOF utilizing smaller seed crystals in a homo-epitaxial growth. The work carried out is interesting, however, it doesn’t give new insight to the studies already carried out in fabricating large crystals of MOF. Major claim in this article is the fabrication of large single crystal Zn-MOF utilizing homoepitaxial growth on a seed crystal. The structure has been confirmed using Single crystal XRD and DFT together. It has been earlier reported for synthesis of few mm sized MOF single crystals [e.g. MOF-5 (some cited by authors) and others] utilizing wet-chemical solvothermal methods for nucleation and aging [Angew. Chem. Int. Ed.2011,50, 276–279, J. Am. Chem. Soc. 2010, 132, 46, 16358–16361, Cell Rep. Phys. Sci. 2022; 3101004, CrystEngComm, 2013,15, 4094-4098, CrystEngComm, 2013,15, 4094-4098, ACS Omega 2021, 6, 27, 17289–17298]. The initial growth of the crystal seed is similar to usually reported MOF single crystal growth utilizing different parameter and solvent, however the homoepitaxial growth is interesting. In this context, the work done doesn’t provide new insights. I recommend to submit to more specific synthesis journals like Crystal growth journals. Apart from this, a few more points I would like the authors to consider which will improve the quality of manuscript:

1. Authors did not convey in the text how their synthesis and growth method is unique and different than others reported already?

Reply:

In section 2.1, the use of DMSO as a solvent for the MOF synthesis is mentioned as a unique method.

In order to highlight this, the following paragraph in the introduction section has been added.

“In this study, we synthesize a MOF built of zinc, acetate and terephthalate ions, that encapsulate arrays of an octahedral zinc dimethyl sulfoxide (DMSO) complex within its one-dimensional (1D) channels by mixing zinc acetate and terephthalic acid in DMSO, a rarely used solvent for the synthesis of MOFs.

Zinc acetate and terephthalic acid in a more common solvent, such as N,N-diethylformamide (DEF) or N,N-dimethylformamide (DMF), are the well-known precursors for the synthesis of MOF-5.

It is demonstrated that these precursors in DMSO lead to the formation of large crystals.”

In addition, the host-guest charge transfer is repeatedly mentioned as the unique mechanism that stabilize the crystal growth in the title, abstract, and conclusion.

Why they choose Zinc salt (Zn(acac)₂) in this case not Zn(NO₃)₂/ ZnCl₂? As the text mostly describes crystal growth, a specific text to describe the same is sought.

Reply:

We choose ZnOAc because it is dissolved well in DMSO. We have not try with Zn(NO₃)₂ or ZnCl₂ as a precursor. Whether or not it leads to the formation of a MOF is interesting, but beyond the scope of this paper.

2. In page 3, authors mentioned, “The continuous growth with no size limits can be attributed to host-guest electrostatic attraction between the anionic framework and the cationic 1D fillers that stabilizes the structure at a molecular level.” What is the biggest crystal size grown in the seed solution? What was the reaction time for the first crystal growth? Please mention.

Reply:

This was only mentioned in the figure caption of Figure 1. In the revised manuscript, section 2.1, Figure 1 is cited and the first crystal growth is explained with the following paragraph:

“Panels a, b, c and d in Figure 1 show photographs of crystals growing in the precursor solution taken sequentially after (a) 6 days (b) 8 days (c) 10 days (d) 13 days of the reaction. The tallest crystal reaches 3.7 mm in length.”

3. Authors’ mentioned “DMSO has been rarely used as a solvent”. As known from literature, DMSO has strong coordinating ability with metal ions and a high boiling solvent and complicate in solvent removal process. However, in this case, it is seen that due to higher solubility of precursors and as depicted by the SCXRD structure Zn(DMSO) does stabilizes the pillar structure. Did the authors’ check for other solvent system in the similar preparation method to grow the crystals? Solvent system is very important in achieving high quality MOF crystals. If DMSO is helping the crystals to grow huge then its benefit should be highlighted in the text.

Reply:

Thank you for pointing this out. Yes, we briefly tried the synthesis with DMF or DEF in similar conditions. In both cases, we obtained powdery products that should be MOF-5 as reported elsewhere. We did not investigate the products in detail as we aimed at large crystal growth.

The following revised paragraph in the introduction section highlights the used of DMSO which distinguishes this work from other previous works.

“In this study, we synthesize a MOF built of zinc, acetate and terephthalate ions, that encapsulate arrays of an octahedral zinc dimethyl sulfoxide (DMSO) complex within its one-dimensional (1D) channels by mixing zinc acetate and terephthalic acid in DMSO, a rarely used solvent for the synthesis of MOFs.

Zinc acetate and terephthalic acid in a more common solvent, such as N,N-diethylformamide (DEF) or N,N-dimethylformamide (DMF), are the well-known precursors for the synthesis of MOF-5.

It is demonstrated that these precursors in DMSO lead to the formation of large crystals.”

4. In the supporting information, how figure S5 and S6 are different? What was influence of air/moisture?

Reply:

The difference is with and without hermetical sealing. “The solution was exposed to air.” has been inserted wherever appropriate.

Did the authors used the same setup for large crystal growth? Please clarify.

Reply:

Yes, we used the same glass vial (20 mL volume and 27 mm diameter) in all steps (both seed and epitaxial growth). This is clarified in S1.3 in the revised SI.

5. Authors mentioned, to avoid nucleation of small crystals, the molar ratio of Zn to TPA was increased from ZnOAc to TPA is (3.0 to 1.0) to 4.0 to 5.0. What was the mechanistic influence of such ratio in term of secondary structure building? Please mention in text.

Reply:

The fact is that the nucleation rate of new crystals was greatly reduced when the molar ratio of Zn to TPA is increased to 4.0-5.0. The crystal growth rate is also reduced, but remains to be positive. On the contrary, pure ZnOAc or pure TPA solution dissolves the MOF as demonstrated in the SI, section S4. It seems apparent that the thermodynamic equilibrium is adjustable by varying the molar ratio of Zn to TPA.

6. What was size of the crystal for SC-XRD shown in figure S2b. Please mention.

Reply:

It was a fragment of size 0.12 x 0.08 x 0.04 mm, cut out of a crystal of millimetre across. Note that our beam size is only 0.1 mm and with such heavily twinned samples, smaller fragments are usually better.

“The crystal dimensions are 0.12 x 0.08 x 0.04 mm.” has been added in the caption of Figure S2b.

7. MOF applications that rely on crystal pore engineering. Do the authors have estimate about the pore size distribution in homoepitaxially grown larger crystal structure?

Reply:

We have not estimated the pore size distribution experimentally.

In response to the reviewer's comment, the structural voids have been estimated and visualized using Olex2, calcvoid (Ref. 6 in the SI).

The cell volume is 16977.940 Å³. The structure calculated with a resolution of 0.2 Å occupies 6342.68 Å³ (37% of the cell volume) and the void 10635.26 Å³ (63%).

Figure S3 (below) in the revised SI displays the structural surfaces with a resolution of 0.2 Å.

[6] Dolomanov, O.V.; Bourhis, L.J.; Gildea, R.J.; Howard, J.A.K.; Puschmann, H., OLEX2: A complete structure solution, refinement and analysis program (2009).

8. What was the basis of the time/days taken for large crystal growth? Was it just random?

Reply:

We exchanged the mother solution every 2-4 days, depending on how crystals grew as the growth rate fluctuated sometimes.

Did the authors washed the crystals in between mother solution change?

Reply:

Yes, the methods applied in this study are as follows.

Upon every solution exchange, the crystal was washed in gently-stirred DMSO. After applying this method, if microcrystals were observed to remain on the crystal surfaces, the crystal was dipped in 0.6 M of ZnAc in DMSO in which the crystal surfaces were

edged away as observed in Figure S7. Finally, the crystal was washed once again in DMSO.

For centimetre-long crystals, microcrystals on the surfaces in the middle section of the crystal could not be removed by applying the aforementioned washing methods. In this case, microcrystals on the surfaces were scraped off using a nail file, followed by the same washing procedure.

In the revised SI, section 1.3, the above washing procedures are described.

9. Authors have utilized homoepitaxial growth to grow larger crystals using a secondary solution on successive interval of time, basically a multi-step nature of the crystal growth. However, the secondary growth may give rise to dislocation or interfacial defects that could be detrimental to the properties of MOF crystals in terms of application. Did the authors do any physical property measurement to check for the same?

Reply:

Thank you for pointing this out. The trace of defects caused by the solvent exchange can be observed in the crystal in Figure e-i, and the crystal in Figure 1k, as linear lines perpendicular to the fast growth direction. Importantly, these defects do not lead to distortions of the overall crystal shape, such as intergrowth of multiple separate crystals.

10. For a homoepitaxial growth, I encourage authors to mention the in-plane orientations of the crystal along with the out-of-plane direction in the figure S2b.

Reply:

In the caption of Figure S2b, the following sentence has been added.

“The (100), (010) and (001) faces are perpendicular to axes a, b and c, respectively.”

11. It is seen in the optical images provided that the opacity of the crystal increases at the core, as the crystal grows to centimeter size (See fig 1). Authors mentioned “also, small crystals grow in the middle section of the crystal surface, which need to be scraped off in order for the crystal to epitaxially grow on the side wall”. How the authors getting rid of the smaller crystals during synthesis? Is it in-situ or they are doing on successive session manually? Please mention.

Reply:

Please refer to our reply to comment 8.

Also, for opto-electronic application, a transparent crystal is sought. How does the inner opaque core impact the application of such crystal in electronic application? Please comment.

Reply:

Its opaque appearance means increased scattering in the respective section of the crystal which may limit their optical applications that require transparency. Also, optical properties of defects, such as luminescence, may cause additional impact. It remains a challenge to mitigate this aging effect towards epitaxial growth of large transparent crystals.

12. How many similar large crystals authors' have grown to check the reproducibility? Do all the crystal grows with same kinetics to similar size?

Reply:

We have grown eight crystals in similar conditions for up to 5 months. Five out of the eight crystals grew over 1cm long.

27th Jul 24

Dear Dr Shiozawa,

Your manuscript titled "Host-guest charge transfer for scalable single crystal epitaxy of a metal-organic framework" has now been seen again by our referees, whose comments appear below. In light of their advice I am delighted to say that we are happy, in principle, to publish a suitably revised version in Communications Materials.

We therefore invite you to edit your manuscript to comply with our journal policies and formatting style in order to maximise the accessibility and therefore the impact of your work.

EDITORIAL REQUESTS

* Your manuscript should comply with our policies and format requirements, detailed in our style and formatting guide (<https://www.nature.com/documents/commsj-phys-style-formatting-guide-accept.pdf>).

* Please edit your manuscript according to the editorial requests in the attached table, and outline revisions made in the right hand column. If you have any questions or concerns about any of our requests, please do not hesitate to contact me. It is important that each request be addressed in order to avoid delays in accepting your manuscript. Please upload the completed table with your manuscript files as a Related Manuscript file.

* The editorial requests table also includes a full list of the files that must be provided upon resubmission. Please upload your files according to this table.

* <***OPTIONAL LIFE SCI REPORTING SUMMARY PARA***> Nature journals require authors of life sciences research papers to include relevant details about several elements of experimental and analytical design in their manuscripts. This initiative aims to improve the transparency of reporting and the reproducibility of published results and is described at: www.nature.com/authors/policies/reporting.pdf. To ensure that your manuscript complies with our policy, please pay close attention to the 'methods' and 'legends' sections of our checklist for authors: Reporting requirements for life sciences research. You may also find the following collection of articles on statistics for biologists helpful: Statistics for Biologists.

* An updated editorial policy checklist that verifies compliance with all required editorial policies must be completed and uploaded with the revised manuscript. All points on the policy checklist must be addressed; if needed, please revise your manuscript in response to these points. Please note that this form is a dynamic 'smart pdf' and must therefore be downloaded and completed in Adobe Reader. Clicking this link will download a zip file containing the pdf.

OPEN ACCESS

Communications Materials is a fully open access journal. Articles are made freely accessible on publication. For further information about article processing charges, open access funding, and advice and support from Nature Research, please visit <https://www.nature.com/commsmat/open-access>

Please use the following link to submit your revised files:

[link redacted]

We hope to hear from you within two weeks; please let us know if the process may take longer.

Best regards,

Natalia Shustova, PhD

Editorial Board Member

Communications Materials

orcid.org/0000-0003-3952-1949

REVIEWERS' COMMENTS:

Reviewer #1 (Remarks to the Author):

I have checked the revised version of the paper. The authors have carefully answered the questions raised by the reviewers and revised the manuscript with much necessary data and comments included. The rewritten contents and discussion of the revised manuscript look much better. Therefore, I think this revised manuscript adequately addressed the reviewers' request. It meets the high standards of Communications Materials and is suitable for publication now. A very nice and exciting work!

Reviewer #2 (Remarks to the Author):

I am grateful for your efforts in revising the manuscript. I believe that the revisions have considerably enhanced the manuscript's clarity and coherence. The manuscript has been well revised and is now suitable for acceptance.

Reviewer #3 (Remarks to the Author):

Authors have addressed the queries.